# Applying Sequential Pattern Mining to Investigate the Temporal Relationships between Commonly Occurring Internal Medicine Diseases and Intervals for the Risk of Concurrent Disease in Canine Patients

**DOI:** 10.3390/ani13213359

**Published:** 2023-10-29

**Authors:** Suk-Jun Lee, Jung-Hyun Kim

**Affiliations:** 1Department of Business Management, Kwangwoon University, 536 Nuri-Hall, 20 Kwangwoon-ro, Nowon-gu, Seoul 01897, Republic of Korea; sjlee@kw.ac.kr; 2Department of Veterinary Internal Medicine, College of Veterinary Medicine, Konkuk University, #120 Neungdong-ro, Gwangjin-gu, Seoul 05029, Republic of Korea

**Keywords:** comorbidities, concurrent diseases, disease intervals, dogs, electric medical record, sequential pattern mining, temporal relationship, veterinary internal medicine diseases

## Abstract

**Simple Summary:**

This study used a technique called sequential pattern mining to uncover connections between common internal medicine diseases in dogs. The goal was to understand how these diseases relate to each other over time. Researchers collected medical records from dogs treated at the Konkuk University Veterinary Medicine Teaching Hospital, focusing on their diseases and the time intervals between diagnoses. They also calculated the 3-year risk of developing another disease after the initial diagnosis. This study identified 547 dogs with at least one internal medicine disease. The sequential pattern mining analysis revealed strong associations and time intervals for five of the most common diseases in dogs, including hyperadrenocorticism, myxomatous mitral valve disease, canine atopic dermatitis, chronic kidney disease, and chronic pancreatitis. This research suggests that sequential pattern mining is a useful tool for understanding disease connections and predicting future health issues in dogs. Veterinarians can use these findings to recommend preventive measures and treatments for dogs at risk of developing additional medical conditions, ultimately improving the care and health of canine patients.

**Abstract:**

Sequential pattern mining (SPM) is a data mining technique used for identifying common association rules in multiple sequential datasets and patterns in ordered events. In this study, we aimed to identify the relationships between commonly occurring internal medicine diseases in canine patients. We obtained medical records of dogs referred to the Konkuk University Veterinary Medicine Teaching Hospital. The data used for SPM included comorbidities and intervals between the diagnoses of internal medicine diseases. Additionally, we estimated the 3-year risk of developing an additional disease after the initial diagnosis of a commonly occurring veterinary internal medicine disease using logistic regression. We identified 547 canine patients diagnosed with ≥ 1 internal medicine disease. The SPM-based analysis assessed comorbidities and intervals for each of the five most common internal medical diseases, including hyperadrenocorticism, myxomatous mitral valve disease, canine atopic dermatitis, chronic kidney disease, and chronic pancreatitis. The highest values of the association rule were 3.01%, 6.02%, 3.9%, 4.1%, and 4.84%, and the shortest intervals were 1.64, 13.14, 5.37, 17.02, and 1.7 days, respectively. This study proposes that SPM is an effective technique for identifying common associations and temporal relationships between internal medicine diseases, and can be used to assess the probability of additional admission due to the development of the subsequent disease that may be diagnosed in canine patients. The results of this study will help veterinarians suggest appropriate preventive measures or other medical treatments for canine patients with medical conditions that have not yet been diagnosed, but are likely to develop in the short term.

## 1. Introduction

The veterinary healthcare system has made considerable progress in transitioning from paper charts to electronic medical records (EMRs) [1]. This transition has resulted in the accumulation of extensive clinical data, which can serve as a valuable resource for enhancing our understanding of current clinical practices and developing decision support systems [1]. Data mining, the process of uncovering hidden knowledge within vast datasets, has found applications in various sectors, including healthcare [2] and the biomedical field [3,4].

Sequential pattern mining (SPM) is a data-mining technique employed to discover common association rules within multiple sequential datasets, and identify patterns in ordered events stored within a database [1,5,6]. Originating in 1995 at IBM’s Almaden Research Center by Agrawal [7], its initial applications were in the retail industry, where it could predict, for instance, that a customer might purchase the sequel to a book shortly after buying the first installment. Beyond retail, SPM was applied as a methodology to investigate relationships among medical events, and develop predictive models for extensive healthcare data in human medicine [1,8]. SPM-derived studies applied to medicine have demonstrated promising outcomes, including disease susceptibility prediction [9,10], improved understanding of disease progression patterns [11,12], identification of revisit patterns [13,14], enhanced pharmacovigilance for medication safety [15,16], and the exploration of relationships between medical conditions [17,18]. Despite several limitations, including variations in data quality, privacy concerns, the complexity of developing predictive models, and ethical considerations regarding data usage and transparency, medical research employing SPM has been actively reported to date [19,20].

In the field of veterinary medicine, SPM-based relationship analyses have not yet been applied. Given the diversity and complexity of veterinary data involving various animal species, the application of SPM within EMRs emerges as an area of significant potential. This approach can predict future disease statuses based on current patient conditions, which is particularly valuable in veterinary medicine with its diverse data sources and patient populations.

In veterinary medicine, it is common for diseases, especially those related to internal medicine, to remain undiagnosed in canines until clear symptoms develop; thus, predicting their development holds significant importance [21]. Moreover, the prompt and precise diagnosis of internal medicine diseases stands as a pivotal factor in managing and preserving the health of veterinary internal medicine patients. The prognosis of many internal medicine diseases can be affected by the presence of concurrent illnesses, and therapeutic medications targeted at specific diseases may exacerbate comorbidities. Therefore, large-scale studies on the relationships between common internal medicine diseases are imperative to empower veterinarians to deliver more precise prognostic information to owners, and guide appropriate treatment strategies. However, the associations between the most common internal medicine diseases in dogs remain inadequately explored.

This population-based study aimed to identify the relationships between commonly occurring internal medical diseases in canine patients through the utilization of SPM and logistic regression analysis.

## 2. Materials and Methods

This section describes the source of data and methodology for the analysis of relationships among internal diseases used in this study.

### 2.1. Canine Patients

This retrospective study included client-owned dogs referred to the Department of Veterinary Internal Medicine at the Veterinary Medicine Teaching Hospital (VMTH), University of Konkuk, Seoul, Republic of Korea. We conducted a thorough search of the EMRs for dogs diagnosed with internal diseases between November 2014 and December 2017. This study was approved by the University of Konkuk Institutional Animal Care and Use Committee (reference no. IACUC 20158). All procedures adhered to the relevant guidelines and regulations, with informed consent acquired from the owners of all participating dogs.

### 2.2. Identification of Internal Medicine Diseases

We obtained the EMRs of dogs referred to the KU-VMTH between 2014 and 2017. Medical records were retrospectively reviewed, and signalments, clinical signs, and diagnostic evaluations were extracted. Patients with previous internal diseases were excluded, and cases with missing or irregular data were excluded from this study. This study selected the five most common veterinary internal medicine diseases, and estimated the 3-year risk of progression to other diseases for each.

### 2.3. SPM

We employed SPM to examine the relationship between diseases caused by time differences, and generalized sequential pattern (GSP) algorithms were used. The algorithms were developed to address sequence mining challenges, and predominantly rely on the a priori (level-wise) approach. Within this level-wise paradigm, an initial step involves identifying all frequently occurring diseases in a systematic manner. This entails enumerating the appearances of individual elements within the medical records. Subsequent to this, comorbidities are refined by excluding infrequent diseases. Consequently, each comorbidity solely encompasses the frequent elements that were initially present. The refined medical records subsequently serve as an input for the algorithm, necessitating a single comprehensive scan of the entire medical record collection. The algorithm conducts several iterations through the records. During the initial iteration, individual items, termed as 1-sequences, are enumerated. Utilizing these frequent items, a collection of potential 2-sequences is constructed, followed by a subsequent iteration to ascertain their prevalence. These recurrent 2-sequences serve as a basis to produce potential 3-sequences. This methodology is perpetuated until no additional recurrent sequences emerge. The algorithm fundamentally comprises two primary steps. In the first step, given the set of frequent sequences $D_{k-1}$ from the (k−1)th iteration, candidates for the subsequent iteration are derived by self-joining $D_{k-1}$. During the step, any sequence with at least one infrequent subsequence is discarded. In the second step, a search strategy based on a hash tree is utilized to ensure efficient support counting. Ultimately, sequences that are not maximally frequent are excised. Figure 1 shows the GSP algorithm used in the study.

The main parameter is the *k*-length sequence, and the number of diseases included in the sequence is denoted by K. A sequence comprising two diseases is referred to as a 2-length sequence. The SPM measures are based on confidence and duration values. Confidence (see Equation (1)) was defined as the conditional probability of a sequential pattern (i.e., disease “A” to disease “B”) [6], and duration (see Equation (2)) was the average occurrence time between the diagnosis of the initial disease and the diagnosis of the subsequent disease. In this analysis, sex, size, analogous classification, and date were used as variables to identify the sequential pattern of the disease using SAS Enterprise Miner statistical software, version 13.2 (SAS Institute, Cary, NC, USA).
(1)$$\mathrm{Confidence}=\frac{P(A\cap B)}{P(A)}$$
(2)$$\mathrm{Duration}=\frac{1}{n}\sum_{i=1}^{n}\mathrm{time}\;\mathrm{of}\;\mathrm{the}\;{\mathrm{pattern}}_{i}$$

### 2.4. Statistical Analysis

To statistically determine whether a specific internal disease occurred more frequently in patients with other commonly occurring internal medicine diseases, we established a cohort of patients diagnosed with internal diseases between 2014 and 2017, excluding those diagnosed before 2014. We compared the logistic regression results with the occurrence of each disease in patients with other newly diagnosed diseases, and the five most common internal medical diseases in the control group. The independent variable was defined as 1 or 0, depending on whether patients newly diagnosed with a disease had another condition, and the covariates were the presence or absence of other diseases. Disease-adjusted odds ratios (aORs) and 95% confidence intervals (CIs) were calculated using multivariate logistic regression analysis. Data analyses were performed using SAS Enterprise Miner, version 13.2 (SAS Institute, Cary, NC, USA). Statistical significance was set at *p* < 0.05.

## 3. Results

This section describes the results of SPM and logistic regression in relation to internal diseases during the study period.

### 3.1. Canine Patient Population

In total, 547 dogs were evaluated for clinical signs of internal diseases during the study period. Overall, 697 diseases were diagnosed, of which, hyperadrenocorticism (HAC), myxomatous mitral valve disease (MMVD), canine atopic dermatitis (CAD), chronic kidney disease (CKD), and chronic pancreatitis were the five most common diseases, accounting for 83.5% (582/697) of all included diagnoses.

Table 1 presents the frequencies stratified by sex, body size, and analogous classification [22] of patients with HAC, MMVD, CAD, CKD, and chronic pancreatitis. The total number of male, castrated male (CM), female, and spayed female (SF) patients was 41, 272, 59, and 210, respectively. Fifty-three breeds of dogs were included: 515 (88.48%) small breeds (weighing 1–9 kg), 60 (10.31%) medium-sized breeds (weighing 10–24 kg), and seven (1.2%) large breeds (weighing 25–54 kg). The median age was 10 years (range, 2 months–19 years); 79 (13.57%) patients were classified as analogous to pediatric patients (younger than 2 years), 346 (59.45%) as adults (3–7 years), 62 (10.65%) as senior (8–10 years), and 95 (16.32%) as geriatric (older than 10 years). HAC was the most commonly diagnosed internal medical disease (152), followed by MMVD (140), CKD (137), CAD (85), and chronic pancreatitis (68). In this study, 189 of the 547 dogs returned to the hospital because of other diseases within 3 years of their initial diagnosis.

### 3.2. Comorbidity Association Rules and Intervals for Internal Medicine Diseases

For each of the five most common internal diseases, the sequential patterns of diseases and intervals between their diagnoses obtained using the SPM are expressed in parentheses, as shown in Figure 2, Figure 3, Figure 4, Figure 5 and Figure 6. The comorbidity association rule “disease A to disease B” indicates the percentage of patients with disease B among the patients with disease A [23]. Duration refers to the interval between the diagnoses of two diseases (average number of days).

#### 3.2.1. Hyperadrenocorticism

Figure 2 illustrates the association rules for HAC and their confidence levels and intervals. The highest value was 3.01% for the following association rules: “HAC to CKD”, “HAC to food allergy”, and “HAC to CAD”. The second highest value was 2.26% for the association rules between HAC and hepatitis, renal calculi, and pyoderma. The association rule for HAC with CKD exhibited the shortest interval of 1.64 days, followed by 2.0, 4.95, 5.36, 6.86, and 9.35 days for the association rules for HAC with hepatitis, renal calculi, CAD, food allergy, and pyoderma, respectively. The highest value of 4.55% was observed for the association rule “hepatitis to HAC”, followed by 4.1%, 3.9%, 3.53%, and 3.33% for associations with CKD, CAD, renal calculi, and food allergies, respectively. The association rule for “CAD to HAC” had the shortest interval of 5.37 days, followed by 6.95, 17.02, 17.35, and 33.78 days for the association rules of food allergy, CKD, renal calculi, and hepatitis with HAC, respectively.

#### 3.2.2. Myxomatous Mitral Valve Disease

Figure 3 shows the association rules for MMVD and their confidence intervals. The highest value was 6.02% for the association rule of “MMVD to HAC”. The association rule for “MMVD to food allergy” had the shortest interval of 13.14 days, followed by 25.22 and 31.67 days for the association rules for MMVD with hepatitis and HAC, respectively. The value of 4.55% was obtained for the association rule of hepatitis and MMVD. The association rule for “food allergy to MMVD” demonstrated the shortest interval of 1.5 days, followed by 50.67 days for the association rule of hepatitis with MMVD.

#### 3.2.3. Chronic Pancreatitis

Figure 4 illustrates the association rules for chronic pancreatitis and their confidence intervals. The association rule for “chronic pancreatitis to HAC” had the highest value of 4.84%. The interval of 1.7 days for the association rule for “chronic pancreatitis to food allergy” was the shortest, followed by 26.63 days for the association rule for chronic pancreatitis and HAC.

#### 3.2.4. Chronic Kidney Disease

Figure 5 shows the association rules for CKD and their confidence intervals. The association rule for “CKD to HAC” had the highest value of 4.1%. Additionally, the association rule “CKD to hypothyroidism” exhibited the shortest interval of 3.1 days, followed by 17.02 days for the association rule of CKD and HAC. The association rule “HAC to CKD” showed the highest value of 3.01%. The association rule for “HAC to CKD” had the shortest interval of 1.64 days, followed by 2.1 days for the association rule for hypothyroidism and CKD.

#### 3.2.5. CAD

Figure 6 displays the association rules for CAD and their confidence intervals. The association rule of “CAD to HAC” had the highest value of 3.9%. The association rule for “CAD to HAC” had the shortest interval of 5.37 days, followed by 7.96 days for the association rule of CAD and pyoderma. The association rule for “pyoderma to CAD” exhibited the highest value of 6.25%, and the association rule for “HAC to CAD” had the shortest interval of 5.35 days, followed by 5.57 days for the association rule for pyoderma and CAD.

### 3.3. Risk of Progression of the Five Most Common Veterinary Internal Medicine Diseases

We estimated the 3-year risk of developing comorbidities in dogs with the five most common veterinary internal medicine diseases using logistic regression analysis. The aORs and CIs for comorbidities are shown in Table 2. Patients with HAC were at an elevated risk of developing CKD and chronic pancreatitis (aOR 1.653 [95% CI 1.086–2.515]; aOR 2.162 [95% CI 1.273–3.672], respectively), whereas patients with MMVD were at a reduced risk of developing CAD and an elevated risk of developing CKD (aOR 0.476 [95% CI 0.255–0.890]; aOR 2.003 [95% CI 1.314–3.051]). Additionally, patients with CKD were at an elevated risk of developing HAC, MMVD, and chronic pancreatitis (aOR 1.582 [95% CI 1.035–2.419]; aOR 2.003 [95% CI 1.314–3.051]; and aOR 3.937 [95% CI 2.318–6.689], respectively). Moreover, patients with chronic pancreatitis had an elevated risk of developing HAC and CKD (aOR, 2.008 [95% CI 1.171–3.444]; aOR, 3.822 [95% CI 2.242–6.513]). However, patients with CAD were at a reduced risk of developing MMVD (aOR, 0.440 [95% CI 0.231–0.837]). No statistically significant decrease or increase in the risk of developing MMVD or CAD was observed in patients with HAC. Further, there was no significant decrease or increase in the risk of developing HAC and chronic pancreatitis in patients with MMVD, the risk of developing HAC, CKD, and chronic pancreatitis in patients with CAD, the risk of developing CAD in patients with CKD, or the risk of developing MMVD and CAD in patients with chronic pancreatitis.

## 4. Discussion

This retrospective study applied SPM to clinical data from canine patients extracted from the EMRs of the VMTH network in the Republic of Korea to assess the probability of additional admission for developing a subsequent disease within 3 years, in patients with at least one veterinary internal medicine disease. We evaluated the comorbidity association rules and intervals among commonly occurring internal medicine diseases in dogs using SPM.

In veterinary medicine, awareness of the possible associations between diseases and the common interval between diagnoses can facilitate early detection [24]. Additionally, some medications used to treat common illnesses can exacerbate other diseases, necessitating careful consideration for patients at high risk for these diseases. Therefore, studies on the comorbidities of commonly occurring internal medicine diseases, and the intervals between diagnoses, are required to inform veterinary clinical practice. Further, when there are many concurrent diseases, confidently linking the reported clinical signs to one specific disease, and not to one or more comorbidities, can be challenging. Moreover, owing to the vague and poorly defined clinical presentation of internal medicine diseases in dogs, identifying the signs that define the clinical presentation of a given disease is difficult. In this study, the shortest interval for disease association was typically less than 1 month. This finding suggests that a concurrent disease may remain undiagnosed until clear symptoms develop, rather than being diagnosed as the disease progresses. Therefore, awareness of the risk of comorbidities in veterinary medicine patients with the most common internal diseases is important. Results of sequence patterns and association mining in this study provided valuable insights into optimizing medical services for disease management, early detection, treatment, and revisits for canine patients with concurrent internal medicine disease patterns. Thus, appropriate preventive measures and recommendations for other medical treatments for canine patients who have been diagnosed with these internal medicine diseases can be provided accordingly.

The percentage of patients with a given disease who experienced the new onset of another specific disease was defined as the confidence parameter [25]. The value of the “confidence” parameter is mathematically synonymous with the concept of “comorbidity” in epidemiology [6]; hence, we regarded the confidence parameter as indicative of comorbidity. We investigated the comorbidities of HAC, MMVD, CAD, CKD, and chronic pancreatitis in canine SPM. We also investigated the interval between the onset of specific diseases.

Regarding the association rules for HAC, the highest value (3.01%) was observed for CKD, food allergies, and CAD. The interval of 1.64 days found for the association of HAC with CKD was the shortest. For the association rules for MMVD, the value of 6.02% for the association between MMVD and HAC was the highest. In this study, although the highest value of the association rule was less than 5%, this result is noteworthy because it indicates that patients with a specific disease had a significantly higher risk of developing another disease than those without a specific disease. Additionally, the association between MMVD and food allergies had the shortest interval of 13.14 days. Regarding the association rules for CAD, the value of 3.9% for the association with HAC was the highest. The association of CAD with HAC showed the shortest interval (5.37 days). Concerning the association rules for CKD, the highest value of 4.1% was obtained for an association with HAC. The association between CKD and HAC exhibited the shortest interval (17.02 days). Finally, for the association rules for chronic pancreatitis, the value of 4.84% for association with HAC was the highest. An interval of 1.7 days for the association of chronic pancreatitis with food allergies was the shortest.

In the present study, many internal medicine diseases were diagnosed in canine patients within a few days of the initial diagnosis. Thus, in patients diagnosed with an internal medicine disease, focusing solely on the initial diagnosis may result in missed opportunities to prevent additional admissions, including those for other internal medicine diseases. It is important to monitor canine patients with the five most common diseases for comorbidities that may put them at risk of additional admissions.

Additionally, we estimated the 3-year risk of developing additional diseases after the initial diagnosis of one of the five most common veterinary internal medicine diseases using logistic regression. We found that patients with HAC had a significantly higher risk of developing CKD and chronic pancreatitis than those without HAC. Patients with MMVD had a significantly higher risk of developing CKD than non-MMVD patients. Moreover, patients with CKD had significantly higher risks of HAC, MMVD, and chronic pancreatitis than non-CKD patients. Patients had a significantly higher risk of developing HAC and CKD than those without chronic pancreatitis. However, patients with CAD had a significantly decreased risk of MMVD, and vice versa. This finding may be attributed to differences in the age at onset of these two diseases: CAD occurs predominantly in relatively young dogs [26], whereas MMVD is usually diagnosed in relatively old dogs [27]. These findings provide valuable insights into the co-occurrence of diseases reported in previous studies on HAC [28], MMVD [29], CKD [30], and chronic pancreatitis [31] in canine patients.

However, given the retrospective nature of this study, there were several inherent limitations, such as incomplete medical records, nonstandardized diagnostic tools, and clinician bias. Moreover, considering the statistical power, we were unable to control the breed because many breeds (53 breeds/547 dogs) were included in this study. Additionally, the findings of this study may not be equally applicable to dogs of all ages because of differences in the common age at the onset of each disease (e.g., atopic dermatitis is diagnosed predominantly in young dogs, whereas heart disease usually occurs in old dogs). To overcome these limitations, including the publication bias from the single-center retrospective study, further larger-scaled and variably controlled (breed, age, and sex) multicenter epidemiological studies are warranted. Furthermore, if a substantial amount of data become available, we intend to conduct research aimed at predicting the onset dates of diseases after the initial diagnosis. This will involve the utilization of deep learning algorithms such as RNN, LSTM, and GRU that can effectively analyze time-series data to enhance our understanding of disease progression.

## 5. Conclusions

In this study, we used SPM to determine the temporal relationships between the onset of internal medicine diseases, visualize these relationships, and generate rules for assessing the probability of additional admission for the development of a subsequent disease that is likely to be diagnosed in canine patients. These findings can provide valuable information to enhance the quality of medical services by recommending suitable medical follow-ups and treatments for the subsequent visits, based on a better understanding of the patterns of concurrent internal medicine diseases in canine patients.

## Figures and Tables

**Figure 1 animals-13-03359-f001:**
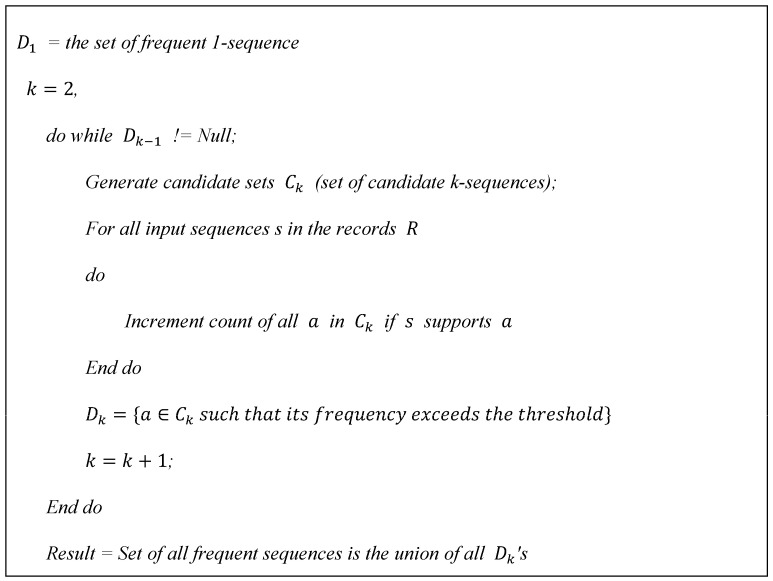
The SPM algorithm used in this study.

**Figure 2 animals-13-03359-f002:**
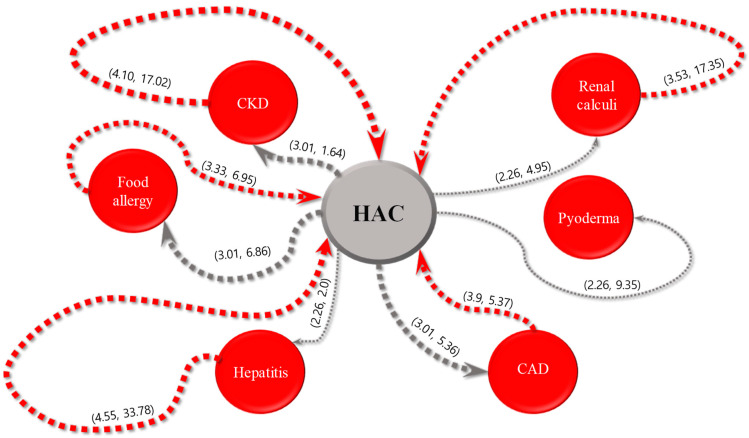
Visualization of the sequential patterns (association rules: comorbidity and average interval between the diagnosis of diseases in days) between hyperadrenocorticism (HAC), chronic kidney disease (CKD), food allergy, hepatitis, renal calculi, pyoderma, and canine atopic dermatitis (CAD).

**Figure 3 animals-13-03359-f003:**
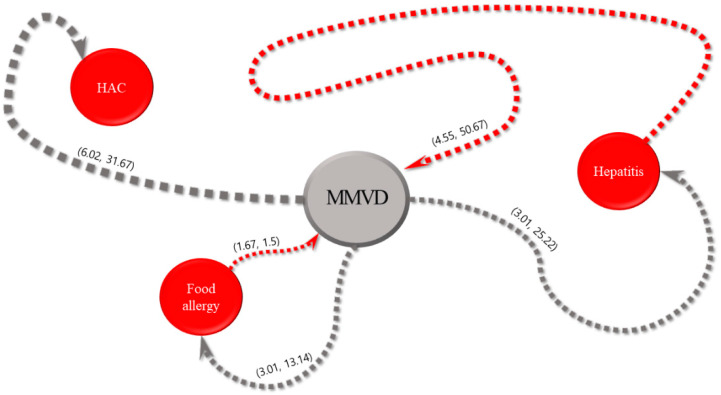
Visualization of sequential patterns between myxomatous mitral valve disease (MMVD), hyperadrenocorticism (HAC), food allergy, and hepatitis.

**Figure 4 animals-13-03359-f004:**
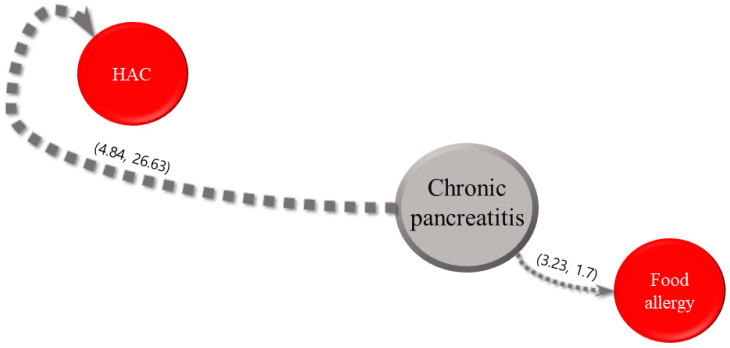
Visualization of sequential patterns between chronic pancreatitis, hyperadrenocorticism (HAC), and food allergy.

**Figure 5 animals-13-03359-f005:**
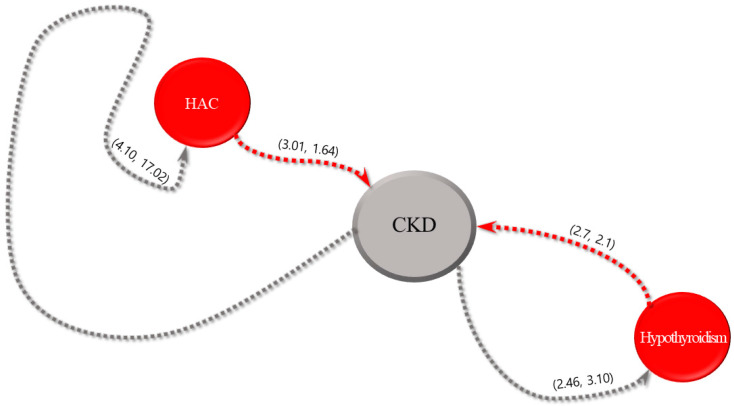
Visualization of sequential patterns between chronic kidney disease (CKD), hyperadrenocorticism (HAC), and hypothyroidism.

**Figure 6 animals-13-03359-f006:**
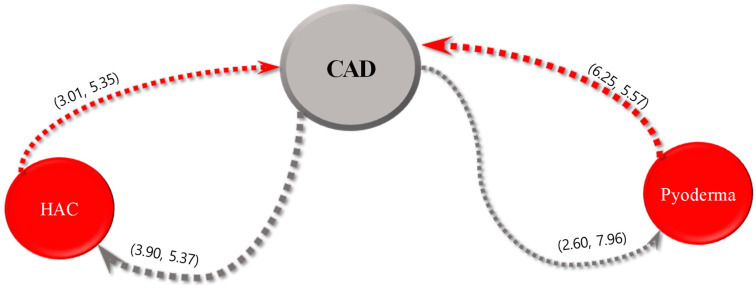
Visualization of sequential patterns between canine atopic dermatitis (CAD), hyperadrenocorticism (HAC), and pyoderma.

**Table 1 animals-13-03359-t001:** Characteristics of the study population with diagnoses of HAC, MMVD, CAD, CKD, and chronic pancreatitis.

Dx	Gender	Frequency	Size ^†^	Frequency	Analogy ^‡^	Frequency	Total
HAC	Female	15	Small	132	Pediatric	11	152
	Spayed female	53	Medium	19	Adult	91	
	Male	10	Large	1	Senior	21	
	Castrated male	74	Extra Large	0	Geriatric	29	
MMVD	Female	10	Small	130	Pediatric	23	140
	Spayed female	51	Medium	10	Adult	84	
	Male	14	Large	0	Senior	11	
	Castrated male	65	Extra Large	0	Geriatric	22	
CAD	Female	7	Small	68	Pediatric	6	85
	Spayed female	29	Medium	14	Adult	52	
	Male	4	Large	3	Senior	10	
	Castrated male	45	Extra Large	0	Geriatric	17	
CKD	Female	19	Small	123	Pediatric	26	137
	Spayed female	52	Medium	12	Adult	79	
	Male	8	Large	2	Senior	13	
	Castrated male	58	Extra Large	0	Geriatric	19	
Chronic pancreatitis	Female	8	Small	62	Pediatric	13	68
	Spayed female	25	Medium	5	Adult	40	
	Male	5	Large	1	Senior	7	
	Castrated male	30	Extra Large	0	Geriatric	8	

Dx, diagnosis; HAC, hyperadrenocorticism; MMVD, myxomatous mitral valve disease; CAD, canine atopic dermatitis; CKD, chronic kidney disease. † Small, 1–9 kg; medium, 10–24 kg; large, 25–54 kg; extra-large, >54 kg. ‡ Pediatric, younger than 2 years old; adult, 3–7 years old; senior, 8–10 years old; geriatric, older than 10 years old.

**Table 2 animals-13-03359-t002:** Risk of developing an additional disease after the diagnosis of a commonly occurring internal medical disease in 547 dogs.

	Risk of Disease	OR (95% CI)	*p*		Risk of Disease	OR (95% CI)	*p*
HAC ≥ MMVD				CAD ≥ CKD			
Controls	0.237 (94/396)	Reference		Controls	0.178 (15/84)	Reference	
HAC	0.286 (42/147)	1.285 (0.839–1.968)	0.249	CAD	0.262 (121/462)	0.613 (0.338–1.111)	0.107
HAC ≥ CAD				CAD ≥ Chronic pancreatitis			
Controls	0.134 (53/396)	Reference		Controls	0.118 (10/85)	Reference	
HAC	0.201 (30/149)	1.632 (0.996–2.667)	1.632	CAD	0.123 (57/462)	0.947 (0.463–1.938)	0.882
HAC ≥ CKD				CKD ≥ HAC			
Controls	0.096 (38/396)	Reference		Controls	0.246 (101/410)	Reference	
HAC	0.095 (14/148)	1.653 (1.086–2.515)	0.019 *	CKD	0.341 (45/132)	1.582 (1.035–2.419)	0.034 *
HAC ≥ Chronic pancreatitis				CKD ≥ MMVD			
Controls	0.096 (38/396)	Reference		Controls	0.219 (90/410)	Reference	
HAC	0.187 (28/150)	2.162 (1.273–3.672)	0.004 *	MMVD	0.360 (49/136)	2.003 (1.314–3.051)	<0.001 *
MMVD ≥ CAD				CKD ≥ CAD			
Controls	0.093 (13/140)	Reference		Controls	0.110 (15/136)	Reference	
MMVD	0.177 (72/407)	0.476 (0.255–0.890)	0.02 *	CKD	0.168 (69/410)	0.613 (0.338–1.111)	0.107
MMVD ≥ CKD				CKD ≥ Chronic pancreatitis			
Controls	0.214 (87/407)	Reference		Controls	0.078 (32/410)	Reference	
CKD	0.352 (49/139)	2.003 (1.314–3.051)	<0.001 *	CKD	0.250 (34/136)	3.937 (2.318–6.689)	<0.0001 *
MMVD ≥ Chronic pancreatitis				Chronic pancreatitis ≥ HAC			
Controls	0.115 (16/139)	Reference		Controls	0.254 (122/480)	Reference	
MMVD	0.123 (50/407)	0.929 (0.510–1.691)	0.809	Chronic pancreatitis	0.406 (26/64)	2.008 (1.171–3.444)	0.011 *
MMVD ≥ HAC				Chronic pancreatitis ≥ MMVD			
Controls	0.258 (105/407)	Reference		Controls	0.231 (15/65)	Reference	
MMVD	0.314 (43/137)	1.316 (0.861–2.010)	0.204	Chronic pancreatitis	0.256 (123/480)	0.871 (0.472–1.606)	0.658
CAD ≥ HAC				Chronic pancreatitis ≥ CAD			
Controls	0.257 (119/462)	Reference		Controls	0.149 (10/67)	Reference	
CAD	0.354 (29/82)	1.577 (0.958–2.596)	0.073	Chronic pancreatitis	0.156 (75/480)	0.947 (0.463–1.938)	0.882
CAD ≥ MMVD				Chronic pancreatitis ≥ CKD			
Controls	0.143 (12/84)	Reference		Controls	0.212 (102/480)	Reference	
CAD	0.275 (127/462)	0.440 (0.231–0.837)	0.012 *	Chronic pancreatitis	0.508 (33/65)	3.822 (2.242–6.513)	<0.0001 *

* *p* < 0.05. HAC, hyperadrenocorticism; MMVD, myxomatous mitral valve disease; CKD, chronic kidney disease.

## Data Availability

Data supporting the findings of this study are available from the corresponding author upon request.

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
