# Peer review of "Applying Sequential Pattern Mining to Investigate the Temporal Relationships between Commonly Occurring Internal Medicine Diseases and Intervals for the Risk of Concurrent Disease in Canine Patients"

_animals, 2023, doi:10.3390/ani13213359_

Round 1

Reviewer 1 Report

Comments and Suggestions for Authors

1. While the study provides information on SPM and its application, it lacks a clear explanation of why identifying associations and temporal relationships between internal medicine diseases in dogs is important. How does this contribute to veterinary medicine or animal health? What practical implications can arise from these findings?

2. The study mentions obtaining medical records from a specific hospital. Details regarding the data source, sampling process, inclusion/exclusion criteria, and representativeness of the data are crucial. Are these records from a specific period, region, or type of cases? How do these records reflect the broader population of canine patients?

3. The study uses "comorbidities and intervals between diagnoses" for SPM. It's important to detail the preprocessing steps for this data. How were comorbidities defined and captured? How were intervals calculated, and were any considerations made for missing or irregular data?

4. The study describes the use of logistic regression for comparing disease occurrence between patients with different diagnoses. More details are needed regarding the independent variables included in the model. How were these variables selected, and were potential confounding factors considered?

4. The study presents the highest values of association rules and shortest intervals for the five most common internal medical diseases. It's essential to provide more context for these findings. How were these values calculated? What do they signify in terms of disease relationships? Were these values compared to any baseline or statistical significance?

5. The study focuses on one specific hospital's data. How generalizable are these findings to other veterinary hospitals or different regions? Can the relationships and patterns found here be considered representative of broader canine populations?

6. The study should conclude with a discussion of the clinical relevance of its findings. How can veterinarians and animal health professionals benefit from these insights? Are there potential applications in disease management, early detection, or treatment planning?

7. It's important to acknowledge any limitations of the study, such as data quality, scope, and assumptions made. Also, suggest potential areas for future research or improvements in methodology.

8. More references on data mining in biomedical studies should be added to attract a broader readership i.e., PMID: 37120403, PMID: 37112302.

9. The study focuses on dogs referred to a specific veterinary hospital within a specific time frame. It's important to discuss how representative this sample is of the broader population of canine patients. Are there any potential biases introduced due to the selection criteria or time frame?

10. The study excludes patients with previous internal diseases. The rationale behind this exclusion should be provided. How might excluding such cases impact the understanding of disease progression and relationships?

11. The study mentions using sex, size, analogous classification, and date as variables for identifying sequential patterns. The choice of these variables should be justified. Are there other variables that could potentially impact disease progression but were not considered?

12. The methodology briefly mentions using SAS Enterprise Miner for SPM, but it lacks details about the specific SPM algorithm used, parameters chosen, and how confidence and duration thresholds were determined.

13. The study mentions comparing results to the five most common internal medical diseases in the control group. However, it doesn't explain how this control group was selected or how it represents a valid comparison.

14. The authors used multivariate logistic regression for calculating disease-adjusted odds ratios (aORs) and confidence intervals. The specific covariates adjusted for in the model need to be explicitly stated, along with their rationale.

15. It's important to address potential multiple comparisons in statistical significance. Were any corrections applied for multiple hypothesis testing? This is particularly important if multiple comparisons are being made.

16. The authors don't explicitly discuss how the identified sequential patterns and associations will be interpreted in the context of veterinary medicine. How will these findings contribute to understanding disease progression and management?

17. External validation of the results, such as testing the identified associations and patterns in an independent dataset or across different veterinary hospitals, would add strength to the study's conclusions.

18. To ensure research transparency and reproducibility, the authors should consider providing a more detailed methodology, including parameter values, formulas for calculations, and potential code snippets used for analyses.

Comments on the Quality of English Language

English writing and presentation style should be improved.

Author Response

Response to Reviewer 1’ Comments

We would like to thank the reviewers for their constructive comments and suggestions, which have greatly helped us improve the manuscript. The manuscript has been rechecked and the necessary changes have been made in accordance with the reviewers’ suggestions. The responses to all comments have been prepared and provided below. All the corrections in the revised manuscript are highlighted in yellow and line numbers are indicated.

Q1. While the study provides information on SPM and its application, it lacks a clear explanation of why identifying associations and temporal relationships between internal medicine diseases in dogs is important. How does this contribute to veterinary medicine or animal health? What practical implications can arise from these findings?

Response) Since animals cannot communicate with veterinarians or owners, symptoms of the disease are often discovered late, making early diagnosis of the disease difficult. The SPM in this study is expected to enable early diagnosis and treatment of concurrent diseases by predicting the pattern of occurrence intervals of concurrent diseases in common medical diseases in dogs. Therefore, the relevant content was added to the text as follows.

Line 13-24 (Simple Summary): This study used a technique called sequential pattern mining to uncover connections between common internal medicine diseases in dogs. The goal was to understand how these diseases relate to each other over time. Researchers collected medical records from dogs treated at the Konkuk University Veterinary Medicine Teaching Hospital, focusing on their diseases and the time intervals between diagnoses. They also calculated the 3-year risk of developing another disease after the initial diagnosis. The study identified 547 dogs with at least one internal medicine disease. The sequential pattern mining analysis revealed strong associations and time intervals for five of the most common diseases in dogs, including hyperadrenocorticism, myxomatous mitral valve disease, canine atopic dermatitis, chronic kidney disease, and chronic pancreatitis. This research suggests that sequential pattern mining is a useful tool for understanding disease connections and predicting future health issues in dogs. Veterinarians can use these findings to recommend preventive measures and treatments for dogs at risk of developing additional medical conditions, ultimately improving the care and health of canine patients.

Line 36-41: … This study proposes that SPM is an effective technique for identifying common associations and temporal relationships between internal medicine diseases and can be used to assess the probability of additional admission due to the development of the subsequent disease that may be diagnosed in canine patients. The results of this study will help veterinarians suggest appropriate preventive measures or other medical treatments for canine patients with medical conditions that have not yet been diagnosed but are likely to develop in the short term.

Line 57-69: …Additionally, SPM is useful for predicting complications when multiple diseases are present in a patient’s medical records in human medicine [1]. Medicine applications were proposed soon after the development of this technique [6] and eventually facilitated the prediction of disease susceptibility [7,8], disease progression [24,27], revisit patterns [9,23], improvements in pharmacovigilance [10,11], and to investigate medical diseases relationships [25,26]. However, SPM-based relationship analyses have not yet been applied in veterinary medicine.

             In veterinary medicine, it is common for diseases, especially those related to internal medicine, to remain undiagnosed in canines until clear symptoms develop; predicting their development is therefore important [12]. Moreover, prompt and accurate diagnosis of internal medicine diseases is the most critical factor for the management and survival of veterinary internal medicine patients. The prognosis of many internal medicine diseases can be affected by the presence of concurrent illnesses, and therapeutic medications for specific diseases can exacerbate comorbidities.

Line 250-268: In veterinary medicine, awareness of the possible associations between diseases and the common interval between diagnoses can facilitate early detection [15]. Additionally, some medications used to treat common illnesses can exacerbate other diseases, necessitating careful consideration for patients at high risk for these diseases. Therefore, studies on the comorbidities of commonly occurring internal medicine diseases and the intervals between diagnoses are required to inform veterinary clinical practice. Further, when there are many concurrent diseases, confidently linking the reported clinical signs to one specific disease and not to one or more comorbidities can be challenging. Moreover, owing to the vague and poorly defined clinical presentation of internal medicine diseases in dogs, identifying the signs that define the clinical presentation of a given disease is difficult. In this study, the shortest interval for disease association was typically less than 1 month. This finding suggests that a concurrent disease may remain undiagnosed until clear symptoms develop, rather than being diagnosed as the disease progresses. Therefore, awareness of the risk of comorbidities in veterinary medicine patients with the most common internal diseases is important. Results of sequence patterns and association mining in this study provided valuable insights into optimizing medical services for disease management, early detection, treatment, and revisits for canine patients with concurrent internal medicine disease patterns. Thus, appropriate preventive measures and recommendations for other medical treatments for canine patients who have been diagnosed with these internal medicine diseases can be provided accordingly.

Line 322-328: In this study, we used SPM to determine the temporal relationships between the onset of internal medicine diseases, visualize these relationships, and generate rules for assessing the probability of additional admission for the development of a subsequent disease that is likely to be diagnosed in canine patients. These findings can provide valuable information to enhance the quality of medical services by recommending suitable medical follow-ups and treatments for the subsequent visits, based on a better understanding of the patterns of concurrent internal medicine diseases in canine patients.

Line 385-395 (added references):

  1. Ou-Yang, C.; Wulandari, C.P.; Hariadi, R.A.R.; Wang, H.C.; Chen, C. Applying sequential pattern mining to investigate cerebrovascular health outpatients' re-visit patterns. PeerJ. 2018, 6, e5183. DOI:10.7717/peerj.5183.
  2. Wu, Y.S.; Taniar, D.; Adhinugraha, K.; Wang, C.H.; Pai, T.W. Progression to myocardial infarction short-term death based on interval sequential pattern mining. BMC Cardiovasc. Disord. 2023, 23, 394. DOI:10.1186/s12872-023-03393-7.
  3. Bang, C.H.; Yoon, J.W.; Lee, H.J.; Lee, J.Y.; Park, Y.M.; Lee, S.J.; Lee, J.H. Evaluation of relationships between onychomycosis and vascular diseases using sequential pattern mining. Sci. Rep. 2018, 8, 17840. DOI:10.1038/s41598-018-35909-z.
  4. Han, J.H.; Yoon, J.W.; Yook, H.J.; Bang, C.H.; Chun, J.H.; Lee, J.Y.; Park, Y.M.; Lee, S.J.; Lee, J.H. Evaluation of Atopic Dermatitis and Cutaneous Infectious Disorders Using Sequential Pattern Mining: A Nationwide Population-Based Cohort Study. J. Clin. Med. 2022, 11, 3422. DOI:10.3390/jcm11123422.
  5. Pinaire, J.; Chabert, E.; Azé, J.; Bringay, S.; Landais, P. Sequential Pattern Mining to Predict Medical In-Hospital Mortality from Administrative Data: Application to Acute Coronary Syndrome. J. Healthc. Eng. 2021, 5531807. DOI:10.1155/2021/5531807.

Q2. The study mentions obtaining medical records from a specific hospital. Details regarding the data source, sampling process, inclusion/exclusion criteria, and representativeness of the data are crucial. Are these records from a specific period, region, or type of cases? How do these records reflect the broader population of canine patients?

Response) Details regarding the data source, sampling process, and inclusion/exclusion criteria are mentioned in the materials and methods section below (Line 88-94). And the specific hospital in this study is one of only two large tertiary care institutions in Seoul, the capital city of Korea, and receives many diverse and complex cases. In this study, cases were collected over a 3-year period considering seasonal variability, and all cases diagnosed with one or more internal diseases were included. Therefore, this study is representative of the Korean canine patient population. Considering geographical characteristics, the city was specifically added to the text (Line 80-87). In addition, it was mentioned in the text that additional follow-up research is planned to compensate for the limitations of bias in the results of a single center retrospective study (Line 314-320).

Line 80-87: 2.1. Canine patients

             This retrospective study included client-owned dogs that were referred to the Department of Veterinary Internal Medicine at the Veterinary Medicine Teaching Hospital (VMTH), University of Konkuk, Seoul, South Korea. Electronic medical records were searched for dogs diagnosed with internal diseases between November 2014 and December 2017. This study was approved by the University of Konkuk Institutional Animal Care and Use Committee (reference no. IACUC 20158). All methods in this study were performed in accordance with the relevant guidelines and regulations. Informed consent was obtained from the owners of all dogs included in the study.

Line 88-94: 2.2. Identification of internal medicine diseases

             We obtained electronic medical records of dogs referred to the KU-VMTH between 2014 and 2017. Medical records were retrospectively reviewed, and signalments, clinical signs, and diagnostic evaluations were extracted. Patients with previous internal diseases were excluded, and cases with missing or irregular data were excluded from the study. The five most common veterinary internal medicine diseases were selected, and we estimated the 3-year risk of progression to other diseases for each.

Line 314-320: To overcome these limitations, including the publication bias from the single-center retrospective study, further larger-scaled and variably controlled (breed, age, and sex) multicenter epidemiological studies are warranted. Furthermore, if a substantial amount of data become available, we intend to conduct research aimed at predicting the onset dates of diseases after the initial diagnosis. This will involve the utilization of deep learning algorithms like RNN, LSTM, and GRU that can effectively analyze time-series data to enhance our understanding of disease progression.

Q3. The study uses "comorbidities and intervals between diagnoses" for SPM. It's important to detail the preprocessing steps for this data. How were comorbidities defined and captured? How were intervals calculated, and were any considerations made for missing or irregular data?

Response) “Confidence” is “comorbidity”; and “Duration” is “interval” in this study (Line 270-272). Please refer to Section 3.2 (Line 153-158) for the calculation. And if there was missing or irregular data, the respective cases were removed. The contents were added in the manuscript (Line 91-92).

Line 270-272: The value of the “confidence” parameter is mathematically synonymous with the concept of “comorbidity” in epidemiology [4]; hence, we regarded the confidence parameter as indicative of comorbidity.

Line 153-158:

3.2. Comorbidity association rules and intervals for internal medicine diseases

             For each of the five most common internal diseases, the sequential patterns of diseases and intervals between their diagnoses obtained using the SPM are expressed in parentheses, as shown in Figures 1–5. The comorbidity association rule “disease A to disease B” indicates the percentage of patients with disease B among the patients with disease A [14]. Duration refers to the interval between the diagnoses of two diseases (average number of days).

Line: 91-92: Patients with previous internal diseases were excluded, and cases with missing or irregular data were excluded from the study.

Q4. The study describes the use of logistic regression for comparing disease occurrence between patients with different diagnoses. More details are needed regarding the independent variables included in the model. How were these variables selected, and were potential confounding factors considered?

Response) The independent variable was defined as 1 or 0, depending on whether patients newly diagnosed with a disease had another condition. The contents were written in section 2.4 (Line 115-117). Gender and age were not considered as confounding factors. The limitations of gender and age as confounding factors were written in ‘4. Discussion’ (Line 314-320).

Line 115-117: The independent variable was defined as 1 or 0, depending on whether patients newly diagnosed with a disease had another condition and the covariates were the presence or absence of other diseases.

Line 314-320: To overcome these limitations, including the publication bias from the single-center retrospective study, further larger-scaled and variably controlled (breed, age, and sex) multicenter epidemiological studies are warranted. Furthermore, if a substantial amount of data become available, we intend to conduct research aimed at predicting the onset dates of diseases after the initial diagnosis. This will involve the utilization of deep learning algorithms like RNN, LSTM, and GRU that can effectively analyze time-series data to enhance our understanding of disease progression.

Q4. The study presents the highest values of association rules and shortest intervals for the five most common internal medical diseases. It's essential to provide more context for these findings. How were these values calculated? What do they signify in terms of disease relationships? Were these values compared to any baseline or statistical significance?

Response) We included in the study all internal medicine cases that visited the hospital during the recruitment period (2014 to 2017), and among them, we sequentially selected the five most frequently diagnosed internal diseases and analyzed disease occurrence patterns. And the results (comorbidity) of the SPM and duration were derived from two referenced previous studies (Han et al., 2022; Bang et al., 2018).

Line 389-393:

  1. Bang, C.H.; Yoon, J.W.; Lee, H.J.; Lee, J.Y.; Park, Y.M.; Lee, S.J.; Lee, J.H. Evaluation of relationships between onychomycosis and vascular diseases using sequential pattern mining. Sci. Rep. 2018, 8, 17840. DOI:10.1038/s41598-018-35909-z.
  2. Han, J.H.; Yoon, J.W.; Yook, H.J.; Bang, C.H.; Chun, J.H.; Lee, J.Y.; Park, Y.M.; Lee, S.J.; Lee, J.H. Evaluation of Atopic Dermatitis and Cutaneous Infectious Disorders Using Sequential Pattern Mining: A Nationwide Population-Based Cohort Study. J. Clin. Med. 2022, 11, 3422. DOI:10.3390/jcm11123422.

Additionally, please see the response of Reviewer 1’s Question #1

Q5. The study focuses on one specific hospital's data. How generalizable are these findings to other veterinary hospitals or different regions? Can the relationships and patterns found here be considered representative of broader canine populations?

Response) Please see the response of Reviewer 1’s Question #2

Q6. The study should conclude with a discussion of the clinical relevance of its findings. How can veterinarians and animal health professionals benefit from these insights? Are there potential applications in disease management, early detection, or treatment planning?

Response) Please see the response of Reviewer 1’s Question #1

Q7. It's important to acknowledge any limitations of the study, such as data quality, scope, and assumptions made. Also, suggest potential areas for future research or improvements in methodology.

Response) The following content has been included in ‘4. Discussion’

Line 314-320: To overcome these limitations, including the publication bias from the single-center retrospective study, further larger-scaled and variably controlled (breed, age, and sex) multicenter epidemiological studies are warranted. Furthermore, if a substantial amount of data become available, we intend to conduct research aimed at predicting the onset dates of diseases after the initial diagnosis. This will involve the utilization of deep learning algorithms like RNN, LSTM, and GRU that can effectively analyze time-series data to enhance our understanding of disease progression.

Q8. More references on data mining in biomedical studies should be added to attract a broader readership i.e., PMID: 37120403, PMID: 37112302.

Response) As the reviewer's comments, references on data mining in biomedical studies were added in 1. Introduction (Line 52).

Line 52: ..can be applied to healthcare [2] and the biomedical field [28,29].

Line 399-400 (Added references):

  1. Nguyen, H.S.; Ho, D.K.N.; Nguyen, N.N.; Tran, H.M.; Tam, K.W; Le, N.Q.K. Predicting EGFR Mutation Status in Non–Small Cell Lung Cancer Using Artificial Intelligence: A Systematic Review and Meta-Analysis. Acad. Radiol. 2023, S1076-6332, 00179-4. DOI: 10.1016/j.acra.2023.03.040.
  2. Kha, Q.H.; Le, V.H.; Hung, T.N.K.; Nguyen, N.T.K.; Le, N.Q.K. Development and Validation of an Explainable Machine Learning-Based Prediction Model for Drug–Food Interactions from Chemical Structures. Sensors. 2023, 23, 3962. DOI: 10.3390/s23083962.

Q9. The study focuses on dogs referred to a specific veterinary hospital within a specific time frame. It's important to discuss how representative this sample is of the broader population of canine patients. Are there any potential biases introduced due to the selection criteria or time frame?

Response) Please see the response of Reviewer 1’s Question #2

Q10. The study excludes patients with previous internal diseases. The rationale behind this exclusion should be provided. How might excluding such cases impact the understanding of disease progression and relationships?

Response) We have excluded the information outside the relevant study period or diseases not diagnosed at our hospital to avoid compromising the consistency of medical information.

Q11. The study mentions using sex, size, analogous classification, and date as variables for identifying sequential patterns. The choice of these variables should be justified. Are there other variables that could potentially impact disease progression but were not considered?

Response) There are no variables that could potentially impact the disease progression but were not considered. The corresponding author of this study has a PhD and specialty of veterinary internal medicine (diplomate of Korean Collage of Veterinary Internal Medicine, DKCVIM). We have done sufficient academic review when selecting the variables considered in this study.

Q12. The methodology briefly mentions using SAS Enterprise Miner for SPM, but it lacks details about the specific SPM algorithm used, parameters chosen, and how confidence and duration thresholds were determined.

Response) As the reviewer's comments, the contents were added in the section 2.3 (Line 96-99).

Line 96-99: We employed SPM to examine the relationship between diseases caused by time differences and generalized sequential pattern (GSP) algorithms were used. The main parameter is the k-length sequence, and the number of items included in the sequence is denoted by K. A sequence comprising two items is referred to as a 2-length sequence.

Q13. The study mentions comparing results to the five most common internal medical diseases in the control group. However, it doesn't explain how this control group was selected or how it represents a valid comparison.

Response) We included in the study all internal medicine cases that visited the hospital during the recruitment period (2014 to 2017), and among them, we sequentially selected the five most frequently diagnosed internal diseases and analyzed disease occurrence patterns.

Q14. The authors used multivariate logistic regression for calculating disease-adjusted odds ratios (aORs) and confidence intervals. The specific covariates adjusted for in the model need to be explicitly stated, along with their rationale.

Response) The covariates are the presence or absence of other diseases. The contents were added in section 2.4 (Line 115-117).

Line 115-117: The independent variable was defined as 1 or 0, depending on whether patients newly diagnosed with a disease had another condition and the covariates were the presence or absence of other diseases.

Q15. It's important to address potential multiple comparisons in statistical significance. Were any corrections applied for multiple hypothesis testing? This is particularly important if multiple comparisons are being made.

Response) We are sorry. Due to the insufficient number of data, multiple comparative experiments were not conducted. In future research, we are considering k-fold cross validation.

Q16. The authors don't explicitly discuss how the identified sequential patterns and associations will be interpreted in the context of veterinary medicine. How will these findings contribute to understanding disease progression and management?

Response) Please see the response of Reviewer 1’s Question #1

Q17. External validation of the results, such as testing the identified associations and patterns in an independent dataset or across different veterinary hospitals, would add strength to the study's conclusions.

Response) It was mentioned in the text that additional follow-up research is planned to compensate for the limitations of bias in the results of a single center retrospective study.

Line 314-320: To overcome these limitations, including the publication bias from the single-center retrospective study, further larger-scaled and variably controlled (breed, age, and sex) multicenter epidemiological studies are warranted. Furthermore, if a substantial amount of data become available, we intend to conduct research aimed at predicting the onset dates of diseases after the initial diagnosis. This will involve the utilization of deep learning algorithms like RNN, LSTM, and GRU that can effectively analyze time-series data to enhance our understanding of disease progression.

Q18. To ensure research transparency and reproducibility, the authors should consider providing a more detailed methodology, including parameter values, formulas for calculations, and potential code snippets used for analyses.

Response) As the reviewer's comments, the contents were added in the manuscript.

Reviewer 2 Report

Comments and Suggestions for Authors

There are some key comments to be considered by the authors; a major revision is thus recommended. Please refer to my comments as follows.

Comment 1. Refer to the journal’s template to update the format of the list of references.
Comment 2. Abstract:
(a) Considering “We aimed to identify the relationships between commonly occurring internal medicine diseases.”, and “we estimated the 3-year risk of developing an additional disease…”, update the paper title.
(b) Specify the internal medicine diseases.
Comment 3. Keywords: More terms should be included to reflect the scope of the paper better.
Comment 3. Section 1 Introduction:
(a) Literature review is considered missing. Ensure summarizing the methodology, results, and limitations of the latest existing works (mainly recent five-year publications).
(b) Summarize the research contributions of the paper.
Comment 4. Section 2 Materials and Methods:
(a) Add an introductory paragraph before Subsection 2.1.
(b) Why SPM was used?
(c) The methodology is too short, particularly on SPM.
Comment 5. Section 3 Results:
(a) Add an introductory paragraph before Subsection 3.1.
(b) Grid is missing for Tables. Please refer to the journal’s template.
(c) Enhance the resolutions of all figures. Enlarge the figures to confirm no content is blurred.
(d) In written descriptions, do not use arrows. The style of presentation should be formal.
Comment 6. Comparison between proposed work and existing studies is expected.

Comments on the Quality of English Language

Minor spell check is required.

Author Response

Response to Reviewer 2’ Comments

There are some key comments to be considered by the authors; a major revision is thus recommended. Please refer to my comments as follows.

Response) We would like to thank the reviewers for their constructive comments and suggestions, which have greatly helped us improve the manuscript. The manuscript has been rechecked and the necessary changes have been made in accordance with the reviewers’ suggestions. The responses to all comments have been prepared and provided below. All the corrections in the revised manuscript are highlighted in yellow and line numbers are indicated.

Comment 1. Refer to the journal’s template to update the format of the list of references.

Response) We have updated the format of the list of references based on the journal’s template.

Comment 2. Abstract:
(a) Considering “We aimed to identify the relationships between commonly occurring internal medicine diseases.”, and “we estimated the 3-year risk of developing an additional disease…”, update the paper title.

Response) We have updated the paper title as follows:

Line 1-5:

Applying sequential pattern mining to investigate the temporal relationships between commonly occurring internal medicine diseases and intervals for the risk of concurrent disease in canine patients

(b) Specify the internal medicine diseases.

Response) We have specified the internal medicine diseases as follows:

Line 32-34: The SPM-based analysis of comorbidities and intervals for each of the five most common internal medical diseases, including hyperadrenocorticism, myxomatous mitral valve disease, canine atopic dermatitis, chronic kidney disease, and chronic pancreatitis.

Comment 3. Keywords: More terms should be included to reflect the scope of the paper better.

Response) We have added the more keywords to reflect the scope of the paper better as follows:

Line 43-45 (Keywords): veterinary internal medicine diseases; sequential pattern mining; comorbidities; disease intervals; dogs; temporal relationship; electric medical record; concurrent diseases

Comment 3. Section 1 Introduction:
(a) Literature review is considered missing. Ensure summarizing the methodology, results, and limitations of the latest existing works (mainly recent five-year publications).

Response) We have added medical papers that applied SPM within the last 5 years to the reference list as follows.

Line 385-400 (added references):

  1. Ou-Yang, C.; Wulandari, C.P.; Hariadi, R.A.R.; Wang, H.C.; Chen, C. Applying sequential pattern mining to investigate cerebrovascular health outpatients' re-visit patterns. PeerJ. 2018, 6, e5183. DOI:10.7717/peerj.5183.
  2. Wu, Y.S.; Taniar, D.; Adhinugraha, K.; Wang, C.H.; Pai, T.W. Progression to myocardial infarction short-term death based on interval sequential pattern mining. BMC Cardiovasc. Disord. 2023, 23, 394. DOI:10.1186/s12872-023-03393-7.
  3. Bang, C.H.; Yoon, J.W.; Lee, H.J.; Lee, J.Y.; Park, Y.M.; Lee, S.J.; Lee, J.H. Evaluation of relationships between onychomycosis and vascular diseases using sequential pattern mining. Sci. Rep. 2018, 8, 17840. DOI:10.1038/s41598-018-35909-z.
  4. Han, J.H.; Yoon, J.W.; Yook, H.J.; Bang, C.H.; Chun, J.H.; Lee, J.Y.; Park, Y.M.; Lee, S.J.; Lee, J.H. Evaluation of Atopic Dermatitis and Cutaneous Infectious Disorders Using Sequential Pattern Mining: A Nationwide Population-Based Cohort Study. J. Clin. Med. 2022, 11, 3422. DOI:10.3390/jcm11123422.
  5. Pinaire, J.; Chabert, E.; Azé, J.; Bringay, S.; Landais, P. Sequential Pattern Mining to Predict Medical In-Hospital Mortality from Administrative Data: Application to Acute Coronary Syndrome. J. Healthc. Eng. 2021, 5531807. DOI:10.1155/2021/5531807.
  6. Nguyen, H.S.; Ho, D.K.N.; Nguyen, N.N.; Tran, H.M.; Tam, K.W; Le, N.Q.K. Predicting EGFR Mutation Status in Non–Small Cell Lung Cancer Using Artificial Intelligence: A Systematic Review and Meta-Analysis. Acad. Radiol. 2023, S1076-6332, 00179-4. DOI: 10.1016/j.acra.2023.03.040.
  7. Kha, Q.H.; Le, V.H.; Hung, T.N.K.; Nguyen, N.T.K.; Le, N.Q.K. Development and Validation of an Explainable Machine Learning-Based Prediction Model for Drug–Food Interactions from Chemical Structures. Sensors. 2023, 23, 3962. DOI: 10.3390/s23083962.

(b) Summarize the research contributions of the paper.

Response) Please see the response of Reviewer 1’s Question #1

Comment 4. Section 2 Materials and Methods:

(a) Add an introductory paragraph before Subsection 2.1.

Response) As the reviewer's comments, the introductory paragraph was added in the section 2 (Line 78-79).

Line 78-79: This section describes the source of data and methodology for the analysis of relationships among internal diseases used in this study.

(b) Why SPM was used?

Response) SPM is used for mining sequence patterns from large databases. It is useful for finding all sequence patterns in the data. We used SPM to analyze the pattern in which a disease occurs, and this was informed by two studies (See 1. Introduction).

(c) The methodology is too short, particularly on SPM.

Response) As the reviewer's comments, the contents were added in the section 2.3 (Line 96-99).

Line 96-99: We employed SPM to examine the relationship between diseases caused by time differences and generalized sequential pattern (GSP) algorithms were used. The main parameter is the k-length sequence, and the number of items included in the sequence is denoted by K. A sequence comprising two items is referred to as a 2-length sequence.

Comment 5. Section 3 Results:

(a) Add an introductory paragraph before Subsection 3.1.

Response) As the reviewer's comments, the introductory paragraph was added in the section 3.1 (Line 123-124).

Line 123-124: This section describes the results of SPM and logistic regression in relation to internal diseases during the study period.

(b) Grid is missing for Tables. Please refer to the journal’s template.

Response) As the reviewer's comments, the table was modified.

(c) Enhance the resolutions of all figures. Enlarge the figures to confirm no content is blurred.

Response) As the reviewer's comments, we have modified resolution of all figures.

(d) In written descriptions, do not use arrows. The style of presentation should be formal.

Response) As the reviewer's comments, we have modified all arrows.

Comment 6. Comparison between proposed work and existing studies is expected.

Response) Although there are papers on disease prediction and correlation analysis using SPM in human medicine, but there are no reports yet in veterinary medicine. Therefore, we are mentioned the previous human papers in the text as follows.

Line 36-41: This study proposes that SPM is an effective technique for identifying common associations and temporal relationships between internal medicine diseases and can be used to assess the probability of additional admission due to the development of the subsequent disease that may be diagnosed in canine patients. The results of this study will help veterinarians suggest appropriate preventive measures or other medical treatments for canine patients with medical conditions that have not yet been diagnosed but are likely to develop in the short term.

Line 57-69: …Additionally, SPM is useful for predicting complications when multiple diseases are present in a patient’s medical records in human medicine [1]. Medicine applications were proposed soon after the development of this technique [6] and eventually facilitated the prediction of disease susceptibility [7,8], disease progression [24,27], revisit patterns [9,23], improvements in pharmacovigilance [10,11], and to investigate medical diseases relationships [25,26]. However, SPM-based relationship analyses have not yet been applied in veterinary medicine.

             In veterinary medicine, it is common for diseases, especially those related to internal medicine, to remain undiagnosed in canines until clear symptoms develop; predicting their development is therefore important [12]. Moreover, prompt and accurate diagnosis of internal medicine diseases is the most critical factor for the management and survival of veterinary internal medicine patients. The prognosis of many internal medicine diseases can be affected by the presence of concurrent illnesses, and therapeutic medications for specific diseases can exacerbate comorbidities. …

Line 250-268: In veterinary medicine, In veterinary medicine, awareness of the possible associations between diseases and the common interval between diagnoses can facilitate early detection [15]. Additionally, some medications used to treat common illnesses can exacerbate other diseases, necessitating careful consideration for patients at high risk for these diseases. Therefore, studies on the comorbidities of commonly occurring internal medicine diseases and the intervals between diagnoses are required to inform veterinary clinical practice. Further, when there are many concurrent diseases, confidently linking the reported clinical signs to one specific disease and not to one or more comorbidities can be challenging. Moreover, owing to the vague and poorly defined clinical presentation of internal medicine diseases in dogs, identifying the signs that define the clinical presentation of a given disease is difficult. In this study, the shortest interval for disease association was typically less than 1 month. This finding suggests that a concurrent disease may remain undiagnosed until clear symptoms develop, rather than being diagnosed as the disease progresses. Therefore, awareness of the risk of comorbidities in veterinary medicine patients with the most common internal diseases is important. Results of sequence patterns and association mining in this study provided valuable insights into optimizing medical services for disease management, early detection, treatment, and revisits for canine patients with concurrent internal medicine disease patterns. Thus, appropriate preventive measures and recommendations for other medical treatments for canine patients who have been diagnosed with these internal medicine diseases can be provided accordingly.

Line 322-328: In this study, we used SPM to determine the temporal relationships between the onset of internal medicine diseases, visualize these relationships, and generate rules for assessing the probability of additional admission for the development of a subsequent disease that is likely to be diagnosed in canine patients. These findings can provide valuable information to enhance the quality of medical services by recommending suitable medical follow-ups and treatments for the subsequent visits, based on a better understanding of the patterns of concurrent internal medicine diseases in canine patients.

Reviewer 3 Report

Comments and Suggestions for Authors

This study authored by Lee and Kim, studied the sequential pattern mining of different internal medicine disease in canines. The article is written well and presented clearly. Since, there are a few corrections and clarifications are needed to respond by the authors for improving the articles.

Minor comments

1. Line 21-22: Verify again the values of the association rule and the shortest intervals.
2. Line 124: Table 1: Provide expansion for "Dx".

Major comments:
1. Line 41:  The authors have explained as "Mining sequential pattern" in References 5/6. Is that same for the title?

2. I suggest the authors to update the importance of this study and the approach in the introduction.

3. Line 115: Verify the number of male/castrated values accordingly. Are 41 dogs for male? Are 272 dogs for the castrated male?

4. Line 150: The missing info of association rule for pyoderma, confidence levels and intervals in the paragraph of 3.2.1.

5. Table 2. The dataset and the information are not clear and confusing without the grid, and alignment. I suggest the authors to make a clear presentation like bar chart and reduce the dataset values in the table.

6. The authors must provide more information on sequential pattern mining (SPM) in the method 2.3 section.

Author Response

Response to Reviewer 3’ Comments

This study authored by Lee and Kim, studied the sequential pattern mining of different internal medicine disease in canines. The article is written well and presented clearly. Since, there are a few corrections and clarifications are needed to respond by the authors for improving the articles.

Response) We would like to thank the reviewers for their constructive comments and suggestions, which have greatly helped us improve the manuscript. The manuscript has been rechecked and the necessary changes have been made in accordance with the reviewers’ suggestions. The responses to all comments have been prepared and provided below. All the corrections in the revised manuscript are highlighted in yellow and line numbers are indicated.

Minor comments

1. Line 21-22: Verify again the values of the association rule and the shortest intervals.

Response) As the reviewer's comments, we have verified the values of the association rule and the shortest intervals.

  1. Line 124: Table 1: Provide expansion for "Dx".

Response) We have additionally provided the full name of Dx in the foot note as follows:

Line 147: Dx, diagnosis; HAC, hyperadrenocorticism; MMVD…

Major comments:

  1. Line 41: The authors have explained as "Mining sequential pattern" in References 5/6. Is that same for the title?

Response) It's the same methodology.

  1. I suggest the authors to update the importance of this study and the approach in the introduction.

* Same as the response of Reviewer 1’s Question #1

Response) Since animals cannot communicate with veterinarians or owners, symptoms of the disease are often discovered late, making early diagnosis of the disease difficult. The SPM in this study is expected to enable early diagnosis and treatment of concurrent diseases by predicting the pattern of occurrence intervals of concurrent diseases in common medical diseases in dogs. Therefore, the relevant content was added to the text as follows.

Line 13-24 (Simple Summary): This study used a technique called sequential pattern mining to uncover connections between common internal medicine diseases in dogs. The goal was to understand how these diseases relate to each other over time. Researchers collected medical records from dogs treated at the Konkuk University Veterinary Medicine Teaching Hospital, focusing on their diseases and the time intervals between diagnoses. They also calculated the 3-year risk of developing another disease after the initial diagnosis. The study identified 547 dogs with at least one internal medicine disease. The sequential pattern mining analysis revealed strong associations and time intervals for five of the most common diseases in dogs, including hyperadrenocorticism, myxomatous mitral valve disease, canine atopic dermatitis, chronic kidney disease, and chronic pancreatitis. This research suggests that sequential pattern mining is a useful tool for understanding disease connections and predicting future health issues in dogs. Veterinarians can use these findings to recommend preventive measures and treatments for dogs at risk of developing additional medical conditions, ultimately improving the care and health of canine patients.

Line 36-41: … This study proposes that SPM is an effective technique for identifying common associations and temporal relationships between internal medicine diseases and can be used to assess the probability of additional admission due to the development of the subsequent disease that may be diagnosed in canine patients. The results of this study will help veterinarians suggest appropriate preventive measures or other medical treatments for canine patients with medical conditions that have not yet been diagnosed but are likely to develop in the short term.

Line 57-69: …Additionally, SPM is useful for predicting complications when multiple diseases are present in a patient’s medical records in human medicine [1]. Medicine applications were proposed soon after the development of this technique [6] and eventually facilitated the prediction of disease susceptibility [7,8], disease progression [24,27], revisit patterns [9,23], improvements in pharmacovigilance [10,11], and to investigate medical diseases relationships [25,26]. However, SPM-based relationship analyses have not yet been applied in veterinary medicine.

             In veterinary medicine, it is common for diseases, especially those related to internal medicine, to remain undiagnosed in canines until clear symptoms develop; predicting their development is therefore important [12]. Moreover, prompt and accurate diagnosis of internal medicine diseases is the most critical factor for the management and survival of veterinary internal medicine patients. The prognosis of many internal medicine diseases can be affected by the presence of concurrent illnesses, and therapeutic medications for specific diseases can exacerbate comorbidities.

Line 250-268: In veterinary medicine, awareness of the possible associations between diseases and the common interval between diagnoses can facilitate early detection [15]. Additionally, some medications used to treat common illnesses can exacerbate other diseases, necessitating careful consideration for patients at high risk for these diseases. Therefore, studies on the comorbidities of commonly occurring internal medicine diseases and the intervals between diagnoses are required to inform veterinary clinical practice. Further, when there are many concurrent diseases, confidently linking the reported clinical signs to one specific disease and not to one or more comorbidities can be challenging. Moreover, owing to the vague and poorly defined clinical presentation of internal medicine diseases in dogs, identifying the signs that define the clinical presentation of a given disease is difficult. In this study, the shortest interval for disease association was typically less than 1 month. This finding suggests that a concurrent disease may remain undiagnosed until clear symptoms develop, rather than being diagnosed as the disease progresses. Therefore, awareness of the risk of comorbidities in veterinary medicine patients with the most common internal diseases is important. Results of sequence patterns and association mining in this study provided valuable insights into optimizing medical services for disease management, early detection, treatment, and revisits for canine patients with concurrent internal medicine disease patterns. Thus, appropriate preventive measures and recommendations for other medical treatments for canine patients who have been diagnosed with these internal medicine diseases can be provided accordingly.

Line 322-328: In this study, we used SPM to determine the temporal relationships between the onset of internal medicine diseases, visualize these relationships, and generate rules for assessing the probability of additional admission for the development of a subsequent disease that is likely to be diagnosed in canine patients. These findings can provide valuable information to enhance the quality of medical services by recommending suitable medical follow-ups and treatments for the subsequent visits, based on a better understanding of the patterns of concurrent internal medicine diseases in canine patients.

Line 385-395 (added references):

  1. Ou-Yang, C.; Wulandari, C.P.; Hariadi, R.A.R.; Wang, H.C.; Chen, C. Applying sequential pattern mining to investigate cerebrovascular health outpatients' re-visit patterns. PeerJ. 2018, 6, e5183. DOI:10.7717/peerj.5183.
  2. Wu, Y.S.; Taniar, D.; Adhinugraha, K.; Wang, C.H.; Pai, T.W. Progression to myocardial infarction short-term death based on interval sequential pattern mining. BMC Cardiovasc. Disord. 2023, 23, 394. DOI:10.1186/s12872-023-03393-7.
  3. Bang, C.H.; Yoon, J.W.; Lee, H.J.; Lee, J.Y.; Park, Y.M.; Lee, S.J.; Lee, J.H. Evaluation of relationships between onychomycosis and vascular diseases using sequential pattern mining. Sci. Rep. 2018, 8, 17840. DOI:10.1038/s41598-018-35909-z.
  4. Han, J.H.; Yoon, J.W.; Yook, H.J.; Bang, C.H.; Chun, J.H.; Lee, J.Y.; Park, Y.M.; Lee, S.J.; Lee, J.H. Evaluation of Atopic Dermatitis and Cutaneous Infectious Disorders Using Sequential Pattern Mining: A Nationwide Population-Based Cohort Study. J. Clin. Med. 2022, 11, 3422. DOI:10.3390/jcm11123422.
  5. Pinaire, J.; Chabert, E.; Azé, J.; Bringay, S.; Landais, P. Sequential Pattern Mining to Predict Medical In-Hospital Mortality from Administrative Data: Application to Acute Coronary Syndrome. J. Healthc. Eng. 2021, 5531807. DOI:10.1155/2021/5531807.

  1. Line 115: Verify the number of male/castrated values accordingly. Are 41 dogs for male? Are 272 dogs for the castrated male?

Response) As the reviewer's comments, we have verified the number of male and castrated male values. Number of castrated male is 272, and intact male is 41 dogs.

  1. Line 150: The missing info of association rule for pyoderma, confidence levels and intervals in the paragraph of 3.2.1.

Response) Please check the sentence below (Line 162-166).

“The second highest value was 2.26% for the association rules between HAC and hepatitis, renal calculi, and pyoderma. The association rule for HAC with CKD exhibited the shortest interval of 1.64 days, followed by 2.0, 4.95, 5.36, 6.86, and 9.35 days for the association rules for HAC with hepatitis, renal calculi, CAD, food allergy, and pyoderma, respectively.”

  1. Table 2. The dataset and the information are not clear and confusing without the grid, and alignment. I suggest the authors to make a clear presentation like bar chart and reduce the dataset values in the table.

Response) As the reviewer's comments, the table was modified.

  1. The authors must provide more information on sequential pattern mining (SPM) in the method 2.3 section.

Response) As the reviewer's comments, the contents were added in the section 2.3 (Line  96-99)

Line 96-99: We employed SPM to examine the relationship between diseases caused by time differences and generalized sequential pattern (GSP) algorithms were used. The main parameter is the k-length sequence, and the number of items included in the sequence is denoted by K. A sequence comprising two items is referred to as a 2-length sequence.

Round 2

Reviewer 1 Report

Comments and Suggestions for Authors

My previous comments have been addressed.

Comments on the Quality of English Language

English writing should be minor checked.

Author Response

Response to Reviewer 1’ Comments

Comment 1) Minor editing of English language required

Response) As per your request, we have made additional edits, including minor grammar and spelling checks. These revisions have been highlighted in blue (grammar revision) within the text. We want to note that we had already obtained professional English proofreading services from a specialized editing company (Editage Corp.; Please see the attached “certificate of editing” file). However, in response to the reviewers' requests, we have included these extra grammar and spelling checks to ensure the highest quality of language in our paper. We hope these revisions meet your expectations, and we appreciate your diligence in reviewing our work.

Reviewer 2 Report

Comments and Suggestions for Authors

The authors have enhanced the quality of the paper. I have some follow-up comment.

Follow-up comments: The literature review is incomplete. Current presentation is limited “Medicine applications were proposed soon after the development of this technique [6] and eventually facilitated the prediction of disease susceptibility [7,8], disease 59 progression [24,27], revisit patterns [9,23], improvements in pharmacovigilance [10,11], and to 60 investigate medical diseases relationships [25,26]. However, SPM-based relationship analyses have not yet been applied in veterinary medicine.
In veterinary medicine, it is common for”. Please summarize the methodology, results and limitations of existing works.

Comment 3 (b) Summarize the research contributions of the paper.
Response) Please see the response of Reviewer 1’s Question #1
Follow-up comment: No research contributions are added in the introduction.

Comment 4. (c) The methodology is too short, particularly on SPM.
Response) As the reviewer's comments, the contents were added in the section 2.3 (Line 96-99).
Follow-up comment: Limited discussion is found in the revision. Please share the figures/tables/equations/pseudo-code where applicable.

Comments on the Quality of English Language

Minor spell check is required.
